# Association of the newly proposed dietary index for gut microbiota and hyperlipidemia: From the 2007–2020 NHANES study

**Fachao Shi[1], Da Yang[1], Quanquan Sun[2], Caoyang Fang [2]***

**1** Department of Cardiology, Maanshan People's Hospital, Maanshan, Anhui, China, **2** Department of Emergency, First Affiliated Hospital of University of Science and Technology of China, Anhui Provincial Hospital, Hefei, Anhui, China

* 3408985159@qq.com

## Abstract

### Objective

The purpose of this study was to investigate the relationship between the Dietary Index for Gut Microbiota (DI-GM) and hyperlipidemia (HL). The DI-GM, a novel index for assessing gut microbiota diversity, has not yet been thoroughly examined in relation to HL.

### Methods

This research involved a cohort of 13,529 individuals enrolled from the National Health and Nutrition Examination Survey (NHANES) between 2007 and 2020. We applied restricted cubic spline (RCS) analysis and weighted multivariable logistic regression to assess the association between DI-GM and HL, supplemented by subgroup analyses to reinforce these findings.

### Results

After multivariable adjustment, subjects with high intake of DI-GM were determined to have a significant reduced risk for developing HL, with a 5% reduced risk for HL for each one standard deviation increased in DI-GM (P = 0.01). In contrast with the group with a DI-GM < 3, HL in the group with a DI-GM > 6 was 40% reduced (P < 0.001). RCS analysis showed a negative linear dose-response relation between DI-GM and development of HL. Subgroup analysis showed an interaction between age-stratification and DI-GM (P = 0.01), but not with gender, racial, BMI, diabetes, and hypertension groups (P > 0.05).

which permits unrestricted use, distribution, and reproduction in any medium, provided the original author and source are credited.

**Data availability statement:** All relevant data are within the paper and its Supporting Information files. Data are also available from the NHANES database [https://wwwn.cdc.gov/nchs/nhanes/default.aspx].

**Funding:** The author(s) received no specific funding for this work.

**Competing interests:** The authors have declared that no competing interests exist.

**Abbreviations:** DI-GM, Dietary index of gut microbiota; RCS, Restricted cubic spline; TG, Triglyceride; TC, Total cholesterol; HDL, High density lipoprotein; LDL, Low density lipoprotein; HbA1c, Glycosylated hemoglobin; BMI, Body mass inde; PIR, Poverty income ratio.

## Conclusion

Our study results show a significant negative linear correlation between DI-GM and HL. However, further research is needed to confirm our findings.

---

## 1. Introduction

The study of the correlation between the DI-GM Aand HL holds significant clinical relevance and necessity, attracting growing interest in the field. Modern lifestyle changes have rendered HL a common metabolic disorder globally. It is not only a crucial risk factor for chronic diseases such as cardiovascular disease and diabetes but is also intimately linked with various metabolic syndromes [1,2]. Consequently, understanding and effectively intervening in HL, particularly through dietary and GM interactions, is of paramount importance.

Recent research indicates that the GM significantly influences the host's metabolic health, especially in lipid metabolism and inflammatory responses. The GM impacts liver lipid synthesis, fatty acid metabolism, and cholesterol absorption and excretion through various mechanisms, thereby regulating blood lipid levels to a certain extent [3,4]. The core of such a regulating role is in its ability to produce short-chain fatty acids (SCFAs) such as acetate and butyrate, not only digestible for intestinal cells, but with an additional impact on the general state of organism's metabolism through systemic circulation, and therefore, improving symptoms in HL patients [5].

DI-GM, as an evaluation tool that integrates dietary structure and GM traits, can become a tool for clinicians to have a deeper insight into GM and its relation with diet. Dietary fiber, prebiotics, and specific types of fermented foods have been shown to effectively stimulate GM diversity and abundance, and in turn, alleviate symptoms of high blood lipid level [6,7]. Clinicians can prescribe individualized intervention diets for high blood lipid level patients through use of the DI-GM index, and in turn, promote intervention effectiveness and alleviate complications such as cardiovascular disease.

Moreover, investigating the mechanism between DI-GM and HL can deepen our knowledge about microorganisms' impact on host metabolism. Not only can it reveal the mechanism through which GM modulates blood lipid, but also provide new prevention and therapeutic approaches for HL. In recent years, studies have confirmed that through nutritional structure adjustment to promote GM composition, not only can blood cholesterol effectively be reduced, but overall metabolic function can also be optimized, and therefore, HL-related disease can be reduced in its risk [8,9].

The application of the DI-GM index in HL patients is of significant clinical value and opens a new path for individualized nutritional intervention. In-depth study in this direction not only provides strong scientific evidence for improving the medical condition of HL patients, but creates a new path for nutrition-microbiology ecolog.

## 2. Materials and methods

### 2.1. Study population

Data were sourced from NHANES, which utilizes a stratified multi-stage sampling technique to collect representative data from the US population, aimed at evaluating

health and nutritional status [10]. Ethical approval was granted by the ethics review committee of the National Center for Health Statistics, with all participants providing written informed consent. The ethical approval requirements for this study were exempted by the hospital ethics committee.

The study analyzed data from 39,950 participants aged 18 years and older from 2007 to 2020. Participants were excluded if they lacked data on high blood lipids (n = 4) or demographic information (n = 9103), such as age and poverty income ratio (PIR), or laboratory information (n = 17314), including blood count and blood lipids. Ultimately, 13,529 participants were included for analysis (Fig 1).

## 2.2. Hyperlipidemia

Hyperlipidemia was identified if participants exhibited any of the following: triglyceride levels ≥150 mg/dL, total cholesterol ≥200 mg/dL, low-density lipoprotein (LDL) levels ≥130 mg/dL, high-density lipoprotein (HDL) levels ≤50 mg/dL for women and ≤40 mg/dL for men [11], or reported using lipid-lowering medications [12,13].

## 2.3. DI-GM

NHANES participants underwent two dietary recall interviews: an initial 24-hour recall conducted at the MEC and a follow-up telephone interview covering detailed dietary intake from the preceding 24 hours. The DI-GM score, as established by Kase et al., incorporates the consumption of 14 types of food or nutrients divided into beneficial and harmful categories. Beneficial items include avocados, broccoli, chickpeas, coffee, cranberries, fermented dairy products, fiber, and whole grains; harmful items include red meat, processed meats, refined grains, and high-fat diets [14]. The scoring

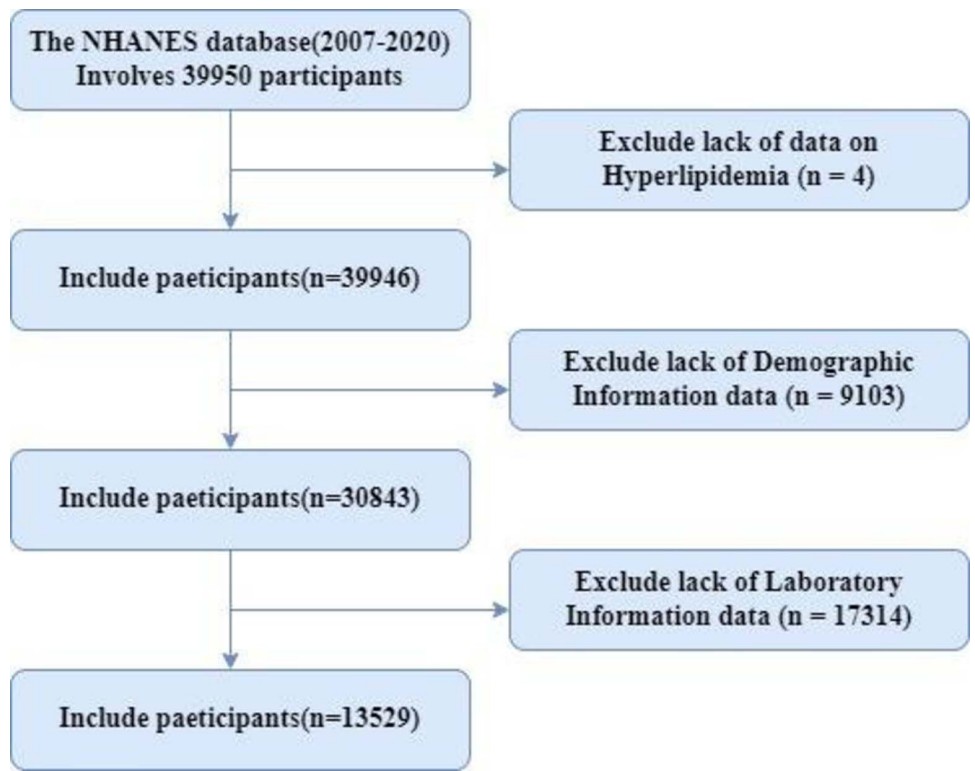

**Fig 1. Study flow chart.**

for DI-GM is derived from NHANES dietary recall data spanning 2007–2020. For beneficial components, a score of 1 is assigned if intake meets or exceeds the gender-specific median, otherwise a score of 0. For harmful components, a score of 1 is assigned if the intake is below the median; otherwise, a score of 0 is given. The total DI-GM score, ranging from 0 to 13, is the sum of all scores (0–9 for beneficial and 0–4 for harmful components). Participants are categorized based on total scores into <3, 3–6, and >6 points [15]. An example clarifies the method: if a participant's diet includes avocados above the median→score 1; red meat below the median→score 1; processed meat above the median→score 0; whole grains below the median→score 1; thus, the total DI-GM score would be 3 points (1 + 1 + 0 + 1).

## 2.4. Covariates

Covariates included demographic and individual data such as household income, categorized into three groups based on the income-to-poverty ratio; smoking status, divided into never, former, or current smokers; and alcohol consumption, categorized into never, former, heavy, moderate, and light drinkers. Hypertension was defined as having systolic blood pressure ≥130mm Hg or diastolic ≥80mm Hg or the use of antihypertensive medication, and diabetes was defined as having been diagnosed by a doctor or the use of glucose-lowering medication.

## 2.5. Statistical analysis

Data analysis was conducted using R software (version 4.3.2), employing the MEC sample weight (WTMEC2YR/7) for weighted analysis. Results for continuous variables were presented as means (SE), and categorical variables as frequencies (%). Multivariate logistic regression models were used to explore the association between DI-GM and HL, with Model 1 unadjusted, Model 2 adjusting for age, race, and gender, and Model 3 further adjusting for diabetes, hypertension, smoking status, alcohol consumption, PIR, triglycerides, total cholesterol, LDL-C, and HDL-C. The RCS method analyzed the relationship, with stratified analyses conducted based on age, gender, race, BMI, hypertension, and diabetes. A p-value <0.05 was considered statistically significant.

## 3. Results

### 3.1. General demographic characteristics of study subjects

This study encompassed 13,529 participants aged 18 or older, with 49.52% being male and 69.60% identifying as non-Hispanic white. Individuals diagnosed with HL were typically older, more likely to be obese, and predominantly non-Hispanic whites, displaying elevated rates of hypertension and diabetes (P<0.05). Notably, those with HL had lower DI-GM scores (P=0.05) (Table 1).

### 3.2. Association between DI-GM and HL

Table 2 details a weighted multivariable logistic regression analysis exploring the association between DI-GM levels and HL. In the unadjusted Model 1, higher DI-GM levels correlated with a reduced incidence of HL. In the fully adjusted Model 3, there was a negative correlation between DI-GM levels and HL, with an OR and 95% CI of 0.95 (0.91, 0.99), signifying a 5% decrease in the likelihood of developing HL for each unit increase in DI-GM. Comparatively, the risk of HL in the DI-GM>6 group was 40% lower than in those with DI-GM<3 (P<0.001), OR (95% CI) being 0.60 (0.45, 0.79). A linear dose-response relationship was delineated by RCS analysis, demonstrating how DI-GM level inversely affects the incidence of HL (Fig 2).

### 3.3. Subgroup analysis

To explore whether factors such as age and BMI, influence the relationship between DI-GM and high triglycerides, a weighted multivariable logistic regression analysis was conducted and the data were stratified accordingly. An interaction

**Table 1. General demographic characteristics of study subjects.**

| Variables | Total | Non-Hyperlipidemia | Hyperlipidemia | P-value |
|---|---|---|---|---|
| **Age, yeaes, mean (SE)** | 47.55(0.28) | 39.82(0.39) | 50.87(0.28) | < 0.0001 |
| **TG, mmol/L, mean (SE)** | 1.29(0.01) | 0.82(0.01) | 1.50(0.01) | < 0.0001 |
| **TC, mmol/L, mean (SE)** | 4.94(0.01) | 4.34(0.01) | 5.20(0.02) | < 0.0001 |
| **HDL, mmol/L, mean (SE)** | 1.41(0.01) | 1.53(0.01) | 1.36(0.01) | < 0.0001 |
| **LDL, mmol/L, mean (SE)** | 2.94(0.01) | 2.43(0.01) | 3.15(0.01) | < 0.0001 |
| **Neutrophils, ×10⁹/L, mean(SE)** | 3.94(0.02) | 3.70(0.04) | 4.04(0.03) | <0.0001 |
| **Lymphocyte, ×10⁹/L, mean(SE)** | 2.00(0.01) | 1.95(0.01) | 2.03(0.01) | <0.0001 |
| **Monocyte, ×10⁹/L, mean(SE)** | 8.23(0.04) | 8.34(0.05) | 8.19(0.05) | 0.02 |
| **Platelet, ×10⁹/L, mean(SE)** | 240.21(0.84) | 232.91(1.25) | 243.34(0.93) | <0.0001 |
| **Creatinine, umol/L, mean (SE)** | 77.61(0.33) | 75.28(0.45) | 78.62(0.42) | < 0.0001 |
| **Uric acid, umol/L, mean (SE)** | 325.78(1.12) | 308.03(1.96) | 333.42(1.24) | < 0.0001 |
| **Blood urea nitrogen, mmol/L, mean (SE)** | 4.91(0.03) | 4.58(0.04) | 5.05(0.03) | < 0.0001 |
| **HbA1c, (%), mean (SE)** | 5.62(0.01) | 5.35(0.01) | 5.74(0.02) | < 0.0001 |
| **DI-GM, mean (SE)** | 5.11(0.03) | 5.18(0.05) | 5.09(0.03) | 0.05 |
| **DI-GM, %(SE)** | | | | 0.04 |
| **<3** | 5.95(0.00) | 5.33(0.47) | 6.22(0.30) | |
| **3-6** | 73.19(0.02) | 72.13(0.99) | 73.65(0.64) | |
| **>6** | 20.86(0.01) | 22.54(0.99) | 20.13(0.68) | |
| **BMI, %(SE)** | | | | <0.0001 |
| **<25** | 29.65(0.01) | 46.28(1.15) | 22.50(0.67) | |
| **25-30** | 33.31(0.01) | 29.41(0.84) | 34.98(0.67) | |
| **>30** | 37.04(0.01) | 24.31(1.06) | 42.52(0.70) | |
| **Sex, %(SE)** | | | | 0.01 |
| Male | 49.52(0.01) | 52.01(1.22) | 48.45(0.56) | |
| Female | 50.48(0.01) | 47.99(1.22) | 51.55(0.56) | |
| **Race, %(SE)** | | | | <0.0001 |
| Mexican American | 8.07(0.01) | 8.36(0.82) | 7.94(0.63) | |
| Non-Hispanic Black | 9.77(0.01) | 12.43(0.90) | 8.62(0.60) | |
| Non-Hispanic White | 69.60(0.03) | 65.70(1.48) | 71.28(1.27) | |
| Other | 12.56(0.01) | 13.52(0.82) | 12.15(0.67) | |
| **Marital, %(SE)** | | | | <0.0001 |
| Married | 63.98(0.02) | 58.35(1.33) | 66.40(0.87) | |
| Never Married | 17.98(0.01) | 28.52(1.13) | 13.45(0.62) | |
| Divorced | 9.16(0.00) | 6.66(0.49) | 10.24(0.51) | |
| Unmarried but have/had partner | 8.88(0.00) | 6.48(0.50) | 9.91(0.45) | |
| **Education, %(SE)** | | | | <0.0001 |
| Less than high School | 13.96(0.01) | 11.30(0.72) | 15.11(0.63) | |
| High school or equivalent | 22.96(0.01) | 21.33(1.03) | 23.67(0.84) | |
| College or above | 63.07(0.02) | 67.37(1.35) | 61.23(1.10) | |
| **Smoke, %(SE)** | | | | <0.0001 |
| Never | 55.68(0.01) | 59.47(1.12) | 54.05(0.87) | |
| Former | 25.80(0.01) | 23.00(1.07) | 27.01(0.78) | |
| Now | 18.52(0.01) | 17.53(0.94) | 18.94(0.67) | |
| **Alcohol, %(SE)** | | | | <0.0001 |
| Never | | | | |

*(Continued)*

**Table 1.** (Continued)

| Variables | Total | Non-Hyperlipidemia | Hyperlipidemia | P-value |
|---|---|---|---|---|
| Former | 11.05(0.01) | 6.66(0.53) | 12.94(0.60) | |
| Mild | 39.44(0.01) | 37.67(1.46) | 40.20(0.90) | |
| Moderate | 18.42(0.01) | 20.31(0.93) | 17.61(0.61) | |
| Heavy | 21.26(0.01) | 25.81(1.20) | 19.30(0.69) | |
| **Diabetes, %(SE)** | | | | <0.0001 |
| Yes | 15.68(0.01) | 6.20(0.47) | 19.76(0.63) | |
| No | 67.09(0.02) | 82.49(0.89) | 60.46(0.81) | |
| Borderline | 17.23(0.01) | 11.32(0.76) | 19.77(0.53) | |
| **Hypertension, %(SE)** | | | | <0.0001 |
| Yes | 37.98(0.01) | 22.63(1.06) | 44.58(0.78) | |
| No | 62.02(0.02) | 77.37(1.06) | 55.42(0.78) | |
| **PIR, %(SE)** | | | | <0.0001 |
| <1.3 | 20.09(0.01) | 21.12(0.99) | 19.65(0.78) | |
| 1.3-3.5 | 35.24(0.01) | 35.03(1.09) | 35.33(0.81) | |
| >3.5 | 44.67(0.02) | 43.85(1.36) | 45.02(1.17) | |

Date are presented as mean (SE) or n (%); TG: Triglyceride, TC: Total cholesterol, HDL: High density lipoprotein, LDL:Low density lipoprotein, HbA1c: Glycosylated hemoglobin, DI-GM: Dietary index for gut microbiota, BMI: Body mass index, PIR: Poverty income ratio.

**Table 2. Association between DI-GM and Hyperlipidemia.**

| Variables | Model 1 | | Model 2 | | Model 3 | |
|---|---|---|---|---|---|---|
| | OR(95%CI) | P | OR(95%CI) | P | OR(95%CI) | P |
| **DI-GM** | 0.97(0.94,1.00) | 0.05 | 0.91(0.89,0.94) | <0.0001 | 0.95(0.91,0.99) | 0.01 |
| **DI-GM** | | | | | | |
| **<3** | Ref | Ref | Ref | Ref | Ref | Ref |
| **3-6** | 0.88(0.72,1.07) | 0.19 | 0.81(0.66,0.98) | 0.03) | 0.79(0.62,1.03) | 0.08 |
| **>6** | 0.77(0.61,0.96) | 0.02 | 0.57(0.46,0.71) | <0.0001 | 0.60(0.45,0.79) | <0.001 |
| **P for trend** | 0.656 | | 0.081 | | 0.661 | |

HR: hazard ratio, CI: confidence interval, Ref: reference.

Model 1: No adjustments made;

Model 2: Adjusted for Age, Sex, Race;

Model 3:Adjusted for Age, Sex, Race, PIR, Hypertension, Diabetes, Smoke, Alcohol, BMI, TG, TC, HDL, LDL.

was observed among age subgroups (P=0.01), but no interactions among gender, race, BMI, diabetes, and hypertension subgroups were found (P>0.05) (Table 3).

## 4. Discussion

This study investigated the relationship between the DI-GM and HL, revealing significant correlations. The study encompassed 13,529 participants, highlighting distinct demographic characteristics in the HL group, particularly among older and obese non-Hispanic whites. These characteristics provide useful baseline information for identifying populations susceptible to HL. Notably, HL patients exhibited a markedly increased prevalence of hypertension and diabetes (P<0.05), aligning with studies linking GM diversity to metabolic state [1,2]. This suggests that HL patients may face multiple metabolic disease risks simultaneously, necessitating systematic evaluation and intervention in clinical settings.

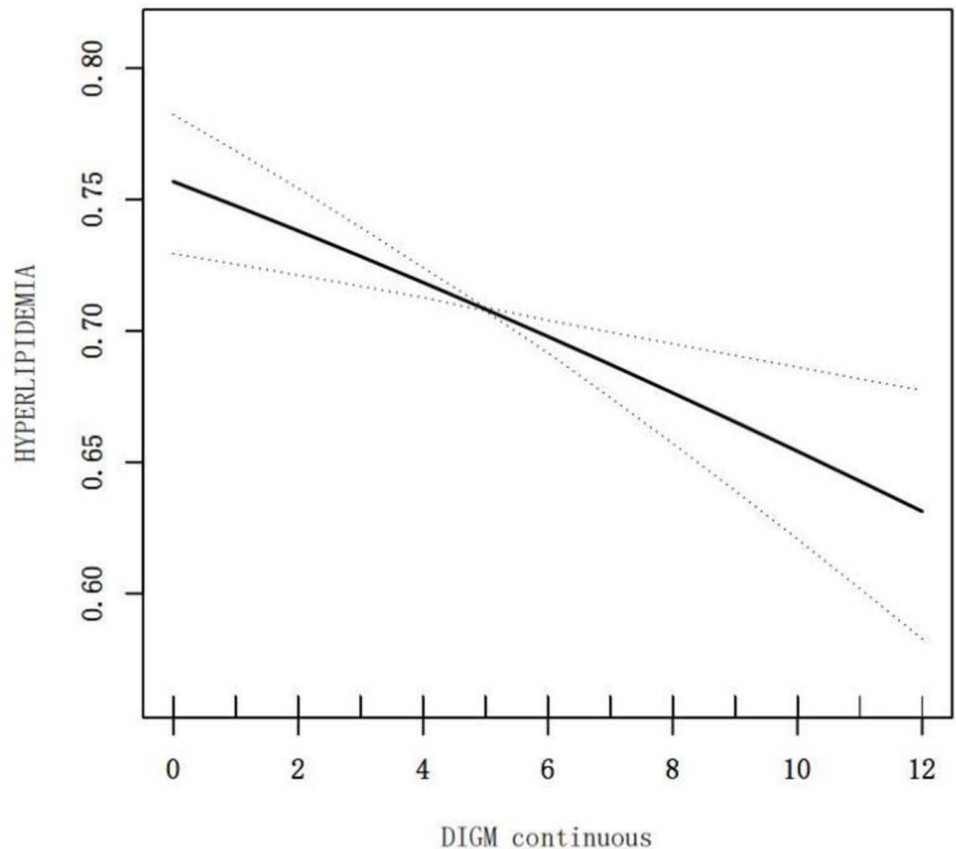

**Fig 2. Analysis of DI-GM association with hyperlipidemia by RCS.** Adjusted for Age, Sex, Race, PIR, Hypertension, Diabetes, Smoke, Alcohol, BMI, TG, TC, HDL, LDL.

The DI-GM is a vital marker of GM diversity and nutritional status, gaining widespread attention in recent years [16,17]. It primarily reflects the balance between GM composition and dietary patterns [18]. In current research, low DI-GM values have been linked to HL, underscoring the regulatory role of GM at a metabolic level [19]. As demonstrated in recent studies, a healthy diet is crucial for maintaining GM balance [20].

Dyslipidemia, commonly characterized by elevated LDL and triglyceride levels and low HDL cholesterol levels [21], is frequently associated with obesity, diabetes, and cardiovascular disease [22]. The GM contributes to the development of these metabolic diseases through various mechanisms, including regulation of inflammatory processes and lipid metabolism [23]. Thus, DI-GM serves as an important tool for assessing GM health and provides new targets for the prevention and treatment of dyslipidemia [24].

The weighted multivariable logistic regression analysis highlighted that an elevation in DI-GM level significantly correlates with a diminished risk of HL. Prior to adjustments, elevated DI-GM levels were consistently linked with a reduced risk of the disease, a trend that persisted in the fully adjusted model. Each incremental unit increase in DI-GM decreased the risk of HL by 5%, underscoring DI-GM's protective effect against HL and its potential to enhance metabolic health through modulation of the gut microbiota [25].

Additionally, compared with the DI-GM < 3 group, the risk of developing HL in the DI-GM > 6 group was significantly reduced by 40% (P < 0.001, OR (95%CI): 0.60(0.45,0.79)). This also supports the usefulness of DI-GM as a prospective

**Table 3. Subgroup analysis.**

| | DI-GM | | | | |
| | Q1 | Q2 | Q3 | p for trend | p for interaction |
|---|---|---|---|---|---|
| **Age** | | | | | 0.01 |
| <60 | Ref | 0.99(0.72,1.37) | 0.85(0.58,1.24) | 0.21 | |
| ≥60 | Ref | 0.33(0.19,0.59) | 0.23(0.12,0.43) | 0.81 | |
| **Sex** | | | | | 0.23 |
| Male | Ref | 0.75(0.50,1.13) | 0.64(0.41,1.00) | 0.43 | |
| Female | Ref | 0.87(0.58,1.30) | 0.57(0.37,0.88) | 0.09 | |
| **Race** | | | | | 0.09 |
| Mexican American | Ref | 0.94(0.57,1.56) | 0.81(0.48,1.36) | 0.34 | |
| Non-Hispanic Black | Ref | 1.05(0.74,1.50) | 0.91(0.56,1.50) | 0.64 | |
| Non-Hispanic White | Ref | 0.71(0.50,1.03) | 0.49(0.33,0.73) | 0.27 | |
| Other | Ref | 0.94(0.49,1.78) | 0.97(0.50,1.89) | 0.78 | |
| **BMI** | | | | | 0.99 |
| <25 | Ref | 0.78(0.56,1.09) | 0.58(0.39,0.88) | 0.03 | |
| 25-30 | Ref | 0.71(0.45,1.11) | 0.49(0.29,0.83) | 0.59 | |
| >30 | Ref | 0.85(0.55,1.30) | 0.72(0.46,1.11) | 0.13 | |
| **Hypertension** | | | | | 0.57 |
| Yes | Ref | 0.84(0.51,1.37) | 0.72(0.43,1.21) | 0.16 | |
| No | Ref | 0.78(0.57,1.07) | 0.56(0.40,0.80) | 0.02 | |
| **Diabetes** | | | | | 0.87 |
| Yes | Ref | 0.73(0.43,1.25) | 0.48(0.25,0.93) | 0.2 | |
| No | Ref | 0.80(0.57,1.12) | 0.63(0.44,0.90) | 0.01 | |
| Borderline | Ref | 0.80(0.41,1.58) | 0.58(0.27,1.24) | 0.18 | |

Adjusted for Age, Sex, Race, PIR, Hypertension, Diabetes, Smoke, Alcohol, BMI, TG, TC, HDL, LDL.

biomarker for the diagnosis of HL risk. In agreement with a large body of literature, where GM diversity and abundance have been strongly associated with a host's metabolism, and have been shown to influence the development of HL by regulating metabolic processes, inflammatory status, and lipid metabolism mechanism [26,27].

In subgroup analyses, factors such as age, gender, race, hypertension, diabetes, and the moderating effects of BMI on the relationship between DI-GM and high triglycerides were evaluated. A significant interaction was found with age (P = 0.01), though no significant interactions were observed concerning gender, race, BMI, diabetes, or hypertension (P > 0.05). That observation concludes that DI-GM may have different mechanism(s) in different age groups [28], and deserves consideration for age in planning intervention measures in an attempt to maximize improvement in high triglyceride risk [29]. It has been previously proven that age can modulate metabolic health through its role in GM composition and function [30].

In this study, a significant correlation between low values of DI-GM and development of HL was noticed. This observation revalidates that GM plays a role in regulating disease in terms of metabolical wellness, in particular, in lipid metabolism. There have been studies supporting that HL is a common metabolical disorder, and it is most frequently accompanied with poor diets [31,32]. Relevant studies refer to that an imbalance in GM can affect the lipid metabolism of a host, subsequently regulating blood concentrations of lipids [33]. For instance, certain beneficial gut microbes can promote lipid metabolism and reduce blood concentrations of lipids through SCFAs production [34]. However, a low value of DI-GM can result in loss of such beneficial metabolites, and subsequently, development of HL [35]. All such observations validate the role of GM in metabolical wellness and require a healthy diet for microbial balance.

This finding is consistent with existing literature and highlights the contribution of a healthy diet to gut microbiome balance. A healthy diet is typically rich in dietary fiber, vitamins, and minerals, and not only contributes to gut microbial diversity but also to enhancing gut barrier function and reducing systemic inflammation [36]. Healthy diets, according to research, can reduce the occurrence of metabolic disease including obesity, cardiovascular disease, and type 2 diabetes [37]. Conversely, low DI-GM is typically associated with high fat and sugar diets and low in fiber, and these types of diets can result in an imbalance in GM and trigger a range of metabolic disease [38]. Specifically, high fat and high sugar diets can trigger pathogen microbes and suppress beneficial microbes, and induce an inflammatory response and increased vulnerability to metabolic disease [35]. Therefore, healthy diets and increasing DI-GM will play a critical role in disease prevention and control of metabolic disease, and will reaffirm the central role of diets in general well-being.

The strength of this study lies in its extensive sample size and comprehensive examination of DI-GM, especially in assessing interactions among various subgroups. Diverging from previous studies that focused solely on individual factors impacting HL, this investigation utilizes a multifactorial approach to illuminate the significant role of DI-GM in the pathogenesis of HL and underscores the necessity for tailored interventions. This study also systematically investigates the dose-response relationship between DI-GM and HL for the first time, providing valuable insights for future clinical strategies. Nonetheless, it has limitations due to its cross-sectional nature, which restricts the ability to establish causality between DI-GM and HL. Prospective studies and randomized controlled trials are essential to confirm these findings and establish a causal relationship. Secondly, the cross-sectional nature makes it challenging to eliminate the influence of unmeasured factors or residual confounding. Thirdly, since this study is based on the American population, generalization of findings to other populations is limited. Further multicenter randomized controlled trials in various countries and regions are needed to verify these findings' generalizability. Fourth, in this study, we did not assess in detail the use of lipid-lowering medications and their separate effect on lipid values in the model. In future research, one might categorize types of lipid-lowering medications and assess their effect on specific lipid markers. That will show more in-depth information on the role of medications in managing dyslipidemia. Fifth, in this study, we did not separate dyslipidemia into its subtypes like hypertriglyceridemia and hypercholesterolemia for detailed analysis. Therefore, in future research, one might assess hypertriglyceridemia and hypercholesterolemia as separate entities in an effort to understand in a better way the patho-logic processes of these two dyslipidemias and their separate association with nutritional factors. Sixth, inasmuch as in this study, a limited set of nutritional factors was not included, in future research, one might combine the use of DI-GM with other nutritional factors in an effort to have a wider overview in terms of findings. That will allow one to understand in a better way the role of different nutritional patterns in managing dyslipidemia.

## 5. Conclusions

In summary, not only is the role of DI-GM in high blood lipid pathogenesis confirmed in this study, but direction for future therapeutic intervention is also gained. Optimising nutritional composition for maximising values of DI-GM can make a contribution towards decreasing high blood lipid and overall metabolic health. This opens the doors for GM-related inter-vention strategies in high blood lipid prevention and therapy.

## Supporting information

**S1 File. Supplementary data.**
(XLSX)

## Author contributions

**Conceptualization:** Fachao Shi, Caoyang Fang.

**Formal analysis:** Caoyang Fang.

**Methodology:** Da Yang, Quanquan Sun.

**Writing – original draft:** Fachao Shi.

**Writing – review & editing:** Fachao Shi.

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
