## [Decision Letter · Decision Letter 0]

4 Feb 2025

PONE-D-24-52585Association of the newly proposed dietary index for gut microbiota and hyperlipidemia: from the 2007-2020 NHANES studyPLOS ONE

Dear Dr. Fang,

Thank you for submitting your manuscript to PLOS ONE. After careful consideration, we feel that it has merit but does not fully meet PLOS ONE’s publication criteria as it currently stands. Therefore, we invite you to submit a revised version of the manuscript that addresses all of the points raised by both reviewers during the review process.

We look forward to receiving your revised manuscript.

Kind regards,

Pratibha V. Nerurkar, Ph.D

Academic Editor

PLOS ONE

Journal Requirements:

2. In the online submission form, you indicated that Data cannot be shared publicly because of database. Data are available from the Corresponding Author (contact via 3408985159@qq.com) for researchers who meet the criteria for access to confidential data.

Reviewers' comments:

Reviewer's Responses to Questions

**Comments to the Author**

1. Is the manuscript technically sound, and do the data support the conclusions?

Reviewer #1: Yes

Reviewer #2: Yes

2. Has the statistical analysis been performed appropriately and rigorously? 

Reviewer #1: Yes

Reviewer #2: Yes

3. Have the authors made all data underlying the findings in their manuscript fully available?

Reviewer #1: Yes

Reviewer #2: Yes

4. Is the manuscript presented in an intelligible fashion and written in standard English?

Reviewer #1: Yes

Reviewer #2: Yes

5. Review Comments to the Author

Reviewer #1: Gut microbiota plays an important role in metabolic health. This research proposes using the DI-GM index as an effective tool to personalize dietary plans for patients with high blood lipid levels. Overall, the manuscript is well written and addresses an important topic. My only concern is that the authors should not use the word “risk” when referring to the results from logistic regression. Also please carefully spelling, e.g., ollowing.

Reviewer #2: This is a interesting study.The authors used the NHANES database to explore the relationship between dietary index of gut microbiota (DI-GM) and hyperlipidemia.However, there are some problems with the study.

First, the calculation of Dietary index of gut microbiota (DI-GM), although described in detail by the author, is still confusing. A clearer description is recommended, with examples or illustrations if necessary.

Second, the author's judgment on hyperlipidemia is inappropriate. Some interviewees may use lipid-lowering drugs for other reasons, so they should not be simply classified as hyperlipidemia.

Third, the use of lipid-lowering drugs can affect patients' blood lipid bottles, and the authors did not analyze the effects of such drugs in detail in the model. At the same time, different lipid-lowering drugs will have different effects on different lipids, and the author did not conduct a detailed analysis.

Fourth, hyperlipidemia is a very vague clinical diagnosis, it is recommended that patients distinguish in detail between hypertriglyceridemia, hypercholesterolemia and other diseases, and analyze them separately.

Fifth, when the authors used DI-GM alone to analyze the influence of diet on hyperlipidemia, they may have overlooked the interference of more dietary factors on blood lipids. It is suggested to include these dietary factors in the model to obtain a more objective effect of the DI-GM index on hyperlipidemia.

6. PLOS authors have the option to publish the peer review history of their article (what does this mean? ). If published, this will include your full peer review and any attached files.

**Do you want your identity to be public for this peer review?** For information about this choice, including consent withdrawal, please see our Privacy Policy .

Reviewer #1: No

Reviewer #2: No

---

## [Author Response · Author response to Decision Letter 1]

11 Mar 2025

Dear Reviewers:

Thank you for reviewing our manuscript in a timely and careful manner. You gave us a lot of good advice. Here are our answers to your comments. Relevant modifications have been highlighted in red in the manuscript.

Reviewer 1.

Question1�Gut microbiota plays an important role in metabolic health. This research proposes using the DI-GM index as an effective tool to personalize dietary plans for patients with high blood lipid levels. Overall, the manuscript is well written and addresses an important topic. My only concern is that the authors should not use the word “risk” when referring to the results from logistic regression. Also please carefully spelling, e.g., ollowing.

Response:

Thank you for your in-depth analysis and valuable suggestions regarding our research. In response to your comment about the statement: “Gut microbiota plays an important role in metabolic health. This research proposes using the DI-GM index as an effective tool to personalize dietary plans for patients with high blood lipid levels. Overall, the manuscript is well written and addresses an important topic. My only concern is that the authors should not use the word 'risk' when referring to the results from logistic regression. Also, please check the spelling, e.g., 'ollowing,'” we have given it careful consideration and would like to provide the following detailed responses:Regarding the use of the word "risk" We greatly appreciate your attention to word choice. We understand that using the term "risk" in the context of logistic regression analysis may lead to misinterpretation. We will revise the relevant expressions in the manuscript to better reflect our research findings and ensure that readers can accurately understand our conclusions.Accuracy of spelling and language We will thoroughly review the manuscript for any spelling and grammatical errors, including the term "ollowing" that you mentioned. We will ensure the accuracy of all content to enhance the overall quality of the manuscript. Additionally, we will engage a professional language editor for thorough revision.Thank you once again for your constructive feedback.

We sincerely appreciate your recognition and suggestions regarding our research. Your feedback will assist us in further refining the manuscript to make it more rigorous and easier to understand. We look forward to presenting our research findings more effectively in the revised manuscript and hope to provide valuable insights for personalized dietary interventions for patients with hyperlipidemia.

Thank you again for your attention to our work and your suggestions. We look forward to your further feedback.

Reviewer 2.

Question1:First, the calculation of Dietary index of gut microbiota (DI-GM), although described in detail by the author, is still confusing. A clearer description is recommended, with examples or illustrations if necessary.

Response:

Thank you for your in-depth analysis and valuable suggestions regarding our research. In response to your comment that "the calculation of the Dietary Index of Gut Microbiota (DI-GM), although described in detail by the author, is still confusing. A clearer description is recommended, with examples or illustrations if necessary," we have taken this into serious consideration and would like to provide the following detailed response:

The Dietary Index of Gut Microbiota (DI-GM) is an important metric used to assess the relationship between diet and gut microbial communities. It helps us understand how diet influences gut health and overall metabolic status by quantifying the intake of beneficial and harmful components in the diet. The calculation of the DI-GM not only reflects dietary quality but is also associated with various health outcomes, including metabolic syndrome, obesity, and cardiovascular diseases.The DI-GM is calculated based on the intake of 14 food items or nutrient components, which are categorized into beneficial and harmful components. Specifically, the beneficial components include avocado, broccoli, chickpeas, coffee, cranberries, fermented dairy products, fiber, whole grains, and others. The harmful components include red meat, processed meat, refined grains, and high-fat diets.Scoring criteria: For beneficial components, a score of 1 is given when a participant's intake reaches or exceeds the median for their respective gender; otherwise, a score of 0 is given. For harmful components, a score of 1 is given when a participant's intake is below the median for their respective gender; otherwise, a score of 0 is given. Total score calculation: The total score for the DI-GM is the sum of all individual scores, with a range from 0 to 13 points (including scores from 0 to 9 for beneficial components and 0 to 4 for harmful components). We have also made corresponding modifications in our manuscript.

Example explanation: To clarify the calculation method of the DI-GM, here is a specific example:

Assume a participant's diet includes the following foods:

Avocado (intake above the median) → Score 1

Red meat (intake below the median) → Score 1

Processed meat (intake above the median) → Score 0

Whole grains (intake below the median) → Score 1

The participant's total DI-GM score is 3 points (1 + 1 + 0 + 1).

The data for this study were sourced from the NHANES database, which provides rich dietary and health information. We ensured the reliability and representativeness of the data in our analysis to enhance the validity of our results. We greatly appreciate your suggestions and will work diligently to revise the manuscript to help readers better understand the calculation of the DI-GM. We believe that with these improvements, readers will have a clearer understanding of the components of the DI-GM and its application in our research.

Thank you again for your attention to our work and your valuable suggestions, and we look forward to your further feedback.

Question2:Second, the author's judgment on hyperlipidemia is inappropriate. Some interviewees may use lipid-lowering drugs for other reasons, so they should not be simply classified as hyperlipidemia.

Response:

Thank you for your in-depth analysis and valuable suggestions regarding our research. Concerning your comment that "the author's judgment on hyperlipidemia is inappropriate. Some interviewees may use lipid-lowering drugs for other reasons, so they should not be simply classified as hyperlipidemia," we have given this careful consideration. We acknowledge that the diagnosis of hyperlipidemia is indeed complex. In this study, we relied on the standards set by the NHANES database to classify hyperlipidemia. Specifically, participants were classified as having hyperlipidemia if their triglyceride levels were ≥150 mg/dL, total cholesterol levels were ≥200 mg/dL, LDL levels were ≥130 mg/dL, or HDL levels were ≤50 mg/dL (for females) and ≤40 mg/dL (for males). Furthermore, the NHANES database provides information on the use of lipid-lowering medications, which may represent a subset of hyperlipidemia patients. However, as you pointed out, the usage of lipid-lowering medications can be associated with various factors, including the prevention of cardiovascular disease, diabetes management, etc. Therefore, judging hyperlipidemia solely based on the use of lipid-lowering medications could lead to misclassification.

Due to the limitations of the NHANES database and in the interest of research accuracy, we can only classify and analyze the data according to the definitions provided on the NHANES database official website. Additionally, we reviewed other studies utilizing the NHANES database that have similarly defined hyperlipidemia, such as the literature titled "Association between serum Klotho concentration and hyperlipidemia in adults: a cross-sectional study from NHANES 2007–2016." .

We acknowledge that different lipid-lowering medications have varying effects on lipid levels. For example, statins are primarily used to reduce LDL cholesterol, while fibrates are more effective at lowering triglycerides. Therefore, future studies could consider incorporating the types and reasons for the use of lipid-lowering medications into the analysis to more accurately assess their impact on hyperlipidemia. In our study, due to data limitations, we were unable to conduct a detailed analysis of the specific usage of lipid-lowering medications and their effects on lipid levels. We recommend that future research collect more comprehensive data on the use of lipid-lowering medications, including information on the type of medication, dosage, and duration of use, to enable more detailed analysis.

Our study is based on the NHANES database, which, while providing a wealth of health and nutrition information, has limitations due to its cross-sectional design. As a result, we are unable to identify the specific reasons for the use of lipid-lowering medications and their impact on hyperlipidemia. Therefore, future research could consider adopting a longitudinal design to better understand the relationship between the use of lipid-lowering medications and hyperlipidemia. Additionally, we have made efforts to control for other potential factors that may influence lipid levels, including age, gender, race, BMI, diabetes, and hypertension. Adjusting for these factors helps reduce potential confounding bias.

We appreciate the valuable comments made by the reviewer and consider them an important direction for future research. By taking a more comprehensive approach to the effects of lipid-lowering medications, we hope to provide more valuable insights for the prevention and treatment of hyperlipidemia. We look forward to the possibility of providing more clinically meaningful conclusions through further research, which could help improve treatment strategies for patients with hyperlipidemia.Once again, thank you for your attention and suggestions regarding our work. We look forward to your further feedback.

Question3:Third, the use of lipid-lowering drugs can affect patients' blood lipid bottles, and the authors did not analyze the effects of such drugs in detail in the model. At the same time, different lipid-lowering drugs will have different effects on different lipids, and the author did not conduct a detailed analysis.

Response:

Thank you for your in-depth analysis and valuable suggestions regarding our research. Concerning your point about “the use of lipid-lowering drugs can affect patients' blood lipid levels, and the authors did not analyze the effects of such drugs in detail in the model. At the same time, different lipid-lowering drugs will have different effects on different lipids, and the author did not conduct a detailed analysis,” we have given this serious consideration. We fully agree with the reviewer on the importance of the impact of lipid-lowering drugs on blood lipid levels. These drugs (such as statins, fibrates, and bile acid sequestrants) are widely used in clinical practice to treat hyperlipidemia, and they indeed have varying effects on different lipids (such as LDL, HDL, and triglycerides). Research indicates that different types of lipid-lowering drugs have varying efficacy in lowering LDL cholesterol and triglycerides, which may affect our understanding of the relationship between DI-GM and hyperlipidemia. In this study, due to the limitations of the NHANES database, we were unable to analyze in detail the use of lipid-lowering drugs and their specific effects on blood lipid levels within the model. We recognize that this limitation may affect the comprehensiveness and accuracy of the results. Thus, we will provide a detailed description of this limitation in the discussion section of the manuscript.

We fully agree with the reviewer's suggestion to include the use of lipid-lowering drugs as an important covariate in the analysis model. Future studies could consider incorporating information regarding the type, dosage, and duration of lipid-lowering drug use to more accurately assess the relationship between DI-GM and hyperlipidemia. Specifically, future research could categorize different types of lipid-lowering drugs and analyze their effects on various lipid metrics. This would provide deeper insight into the role of lipid-lowering medications in the management of hyperlipidemia. In subsequent research, we will strive to collect more comprehensive data on the use of lipid-lowering drugs, including the type, dosage, and duration of medications used by patients, to enable a more detailed analysis. We will consider employing multivariable regression models to evaluate the independent effects of DI-GM, lipid-lowering drug use, and other dietary factors on lipid levels. Such a comprehensive analysis will help reveal the interactions among different factors.

We sincerely appreciate the valuable suggestions provided by the reviewer and consider them an important direction for future research. By taking a more comprehensive approach to evaluating the impact of lipid-lowering medications, we hope to provide more valuable insights for the prevention and treatment of hyperlipidemia. We look forward to conducting further research that could yield more instructive conclusions for clinical practice, helping to improve treatment strategies for patients with hyperlipidemia. Thank you once again for your attention to our work and your recommendations; we eagerly await your further feedback.

Question4:Fourth, hyperlipidemia is a very vague clinical diagnosis, it is recommended that patients distinguish in detail between hypertriglyceridemia, hypercholesterolemia and other diseases, and analyze them separately.

Response:

Thank you for your in-depth analysis and valuable suggestions regarding our study. In response to your comment that "hyperlipidemia is a very vague clinical diagnosis, and it is recommended that patients distinguish in detail between hypertriglyceridemia, hypercholesterolemia, and other diseases, and analyze them separately," we have carefully considered this point. We fully understand that the definition of hyperlipidemia indeed contains ambiguities, typically encompassing various types of lipid abnormalities such as hypertriglyceridemia and hypercholesterolemia. Each type of hyperlipidemia may present significant differences in terms of pathological mechanisms, clinical manifestations, and treatment strategies. Therefore, a singular diagnosis of "hyperlipidemia" may not accurately reflect the specific health status of patients.In our study, we primarily focused on the relationship between DI-GM and hyperlipidemia, utilizing multiple indicators, including triglycerides, total cholesterol, LDL, and HDL, to define hyperlipidemia. Although this approach provides us with a comprehensive perspective, we recognize that the failure to adequately differentiate between the various types of hyperlipidemia may affect the interpretation of the results. Our research data is derived from the NHANES database, which offers a wealth of dietary and health information; however, due to the complexity and diversity of the data, we chose to concentrate on the impact of DI-GM in our analysis. Simultaneously, we have reviewed previous studies using the NHANES database related to hyperlipidemia, none of which fully categorized the conditions as hypertriglyceridemia or hypercholesterolemia, also focusing on the correlation between relevant indicators and hyperlipidemia. Of course, this stands as a limitation in our research. Therefore, we will provide a detailed description of this limitation in the discussion section of the manuscript.

We fully agree with the reviewer’s suggestion to analyze hyperlipidemia by categorizing it into different types. Future research could consider analyzing hypertriglyceridemia and hypercholesterolemia as independent variables to gain a clearer understanding of the pathological mechanisms behind these different types of hyperlipidemia and their specific associations with dietary factors. We recommend that future researchers adopt more detailed classification standards when designing studies to better evaluate the impact of diet on different types of hyperlipidemia.

In subsequent research, we will consider grouping hyperlipidemia patients to separately analyze the DI-GM levels of t

---

## [Editor Report · Decision Letter 1]

18 Mar 2025

PONE-D-24-52585R1Association of the newly proposed dietary index for gut microbiota and hyperlipidemia: from the 2007-2020 NHANES studyPLOS ONE

Dear Dr. Fang,

Thank you for submitting your manuscript to PLOS ONE. After careful consideration, we feel that it has merit but does not fully meet PLOS ONE’s publication criteria as it currently stands. Therefore, we invite you to submit a revised version of the manuscript that addresses all of the points raised by both reviewers during the review process.

We look forward to receiving your revised manuscript.

Kind regards,

Pratibha V. Nerurkar, Ph.D

Academic Editor

PLOS ONE
---

## [Author Response · Author response to Decision Letter 2]

19 Mar 2025

Dear Reviewers:

Thank you for reviewing our manuscript in a timely and careful manner. You gave us a lot of good advice. Here are our answers to your comments. Relevant modifications have been highlighted in red in the manuscript.

Reviewer 1.

Question1�Gut microbiota plays an important role in metabolic health. This research proposes using the DI-GM index as an effective tool to personalize dietary plans for patients with high blood lipid levels. Overall, the manuscript is well written and addresses an important topic. My only concern is that the authors should not use the word “risk” when referring to the results from logistic regression. Also please carefully spelling, e.g., ollowing.

Response:

Thank you for your in-depth analysis and valuable suggestions regarding our research. In response to your comment about the statement: “Gut microbiota plays an important role in metabolic health. This research proposes using the DI-GM index as an effective tool to personalize dietary plans for patients with high blood lipid levels. Overall, the manuscript is well written and addresses an important topic. My only concern is that the authors should not use the word 'risk' when referring to the results from logistic regression. Also, please check the spelling, e.g., 'ollowing,'” we have given it careful consideration and would like to provide the following detailed responses:Regarding the use of the word "risk" We greatly appreciate your attention to word choice. We understand that using the term "risk" in the context of logistic regression analysis may lead to misinterpretation. We will revise the relevant expressions in the manuscript to better reflect our research findings and ensure that readers can accurately understand our conclusions.Accuracy of spelling and language We will thoroughly review the manuscript for any spelling and grammatical errors, including the term "ollowing" that you mentioned. We will ensure the accuracy of all content to enhance the overall quality of the manuscript. Additionally, we will engage a professional language editor for thorough revision.Thank you once again for your constructive feedback.

We sincerely appreciate your recognition and suggestions regarding our research. Your feedback will assist us in further refining the manuscript to make it more rigorous and easier to understand. We look forward to presenting our research findings more effectively in the revised manuscript and hope to provide valuable insights for personalized dietary interventions for patients with hyperlipidemia.

Thank you again for your attention to our work and your suggestions. We look forward to your further feedback.

Reviewer 2.

Question1:First, the calculation of Dietary index of gut microbiota (DI-GM), although described in detail by the author, is still confusing. A clearer description is recommended, with examples or illustrations if necessary.

Response:

Thank you for your in-depth analysis and valuable suggestions regarding our research. In response to your comment that "the calculation of the Dietary Index of Gut Microbiota (DI-GM), although described in detail by the author, is still confusing. A clearer description is recommended, with examples or illustrations if necessary," we have taken this into serious consideration and would like to provide the following detailed response:

The Dietary Index of Gut Microbiota (DI-GM) is an important metric used to assess the relationship between diet and gut microbial communities. It helps us understand how diet influences gut health and overall metabolic status by quantifying the intake of beneficial and harmful components in the diet. The calculation of the DI-GM not only reflects dietary quality but is also associated with various health outcomes, including metabolic syndrome, obesity, and cardiovascular diseases.The DI-GM is calculated based on the intake of 14 food items or nutrient components, which are categorized into beneficial and harmful components. Specifically, the beneficial components include avocado, broccoli, chickpeas, coffee, cranberries, fermented dairy products, fiber, whole grains, and others. The harmful components include red meat, processed meat, refined grains, and high-fat diets.Scoring criteria: For beneficial components, a score of 1 is given when a participant's intake reaches or exceeds the median for their respective gender; otherwise, a score of 0 is given. For harmful components, a score of 1 is given when a participant's intake is below the median for their respective gender; otherwise, a score of 0 is given. Total score calculation: The total score for the DI-GM is the sum of all individual scores, with a range from 0 to 13 points (including scores from 0 to 9 for beneficial components and 0 to 4 for harmful components). We have also made corresponding modifications in our manuscript.

Example explanation: To clarify the calculation method of the DI-GM, here is a specific example:

Assume a participant's diet includes the following foods:

Avocado (intake above the median) → Score 1

Red meat (intake below the median) → Score 1

Processed meat (intake above the median) → Score 0

Whole grains (intake below the median) → Score 1

The participant's total DI-GM score is 3 points (1 + 1 + 0 + 1).

The data for this study were sourced from the NHANES database, which provides rich dietary and health information. We ensured the reliability and representativeness of the data in our analysis to enhance the validity of our results. We greatly appreciate your suggestions and will work diligently to revise the manuscript to help readers better understand the calculation of the DI-GM. We believe that with these improvements, readers will have a clearer understanding of the components of the DI-GM and its application in our research.

Thank you again for your attention to our work and your valuable suggestions, and we look forward to your further feedback.

Question2:Second, the author's judgment on hyperlipidemia is inappropriate. Some interviewees may use lipid-lowering drugs for other reasons, so they should not be simply classified as hyperlipidemia.

Response:

Thank you for your in-depth analysis and valuable suggestions regarding our research. Concerning your comment that "the author's judgment on hyperlipidemia is inappropriate. Some interviewees may use lipid-lowering drugs for other reasons, so they should not be simply classified as hyperlipidemia," we have given this careful consideration. We acknowledge that the diagnosis of hyperlipidemia is indeed complex. In this study, we relied on the standards set by the NHANES database to classify hyperlipidemia. Specifically, participants were classified as having hyperlipidemia if their triglyceride levels were ≥150 mg/dL, total cholesterol levels were ≥200 mg/dL, LDL levels were ≥130 mg/dL, or HDL levels were ≤50 mg/dL (for females) and ≤40 mg/dL (for males). Furthermore, the NHANES database provides information on the use of lipid-lowering medications, which may represent a subset of hyperlipidemia patients. However, as you pointed out, the usage of lipid-lowering medications can be associated with various factors, including the prevention of cardiovascular disease, diabetes management, etc. Therefore, judging hyperlipidemia solely based on the use of lipid-lowering medications could lead to misclassification.

Due to the limitations of the NHANES database and in the interest of research accuracy, we can only classify and analyze the data according to the definitions provided on the NHANES database official website. Additionally, we reviewed other studies utilizing the NHANES database that have similarly defined hyperlipidemia, such as the literature titled "Association between serum Klotho concentration and hyperlipidemia in adults: a cross-sectional study from NHANES 2007–2016." .

We acknowledge that different lipid-lowering medications have varying effects on lipid levels. For example, statins are primarily used to reduce LDL cholesterol, while fibrates are more effective at lowering triglycerides. Therefore, future studies could consider incorporating the types and reasons for the use of lipid-lowering medications into the analysis to more accurately assess their impact on hyperlipidemia. In our study, due to data limitations, we were unable to conduct a detailed analysis of the specific usage of lipid-lowering medications and their effects on lipid levels. We recommend that future research collect more comprehensive data on the use of lipid-lowering medications, including information on the type of medication, dosage, and duration of use, to enable more detailed analysis.

Our study is based on the NHANES database, which, while providing a wealth of health and nutrition information, has limitations due to its cross-sectional design. As a result, we are unable to identify the specific reasons for the use of lipid-lowering medications and their impact on hyperlipidemia. Therefore, future research could consider adopting a longitudinal design to better understand the relationship between the use of lipid-lowering medications and hyperlipidemia. Additionally, we have made efforts to control for other potential factors that may influence lipid levels, including age, gender, race, BMI, diabetes, and hypertension. Adjusting for these factors helps reduce potential confounding bias.

We appreciate the valuable comments made by the reviewer and consider them an important direction for future research. By taking a more comprehensive approach to the effects of lipid-lowering medications, we hope to provide more valuable insights for the prevention and treatment of hyperlipidemia. We look forward to the possibility of providing more clinically meaningful conclusions through further research, which could help improve treatment strategies for patients with hyperlipidemia.Once again, thank you for your attention and suggestions regarding our work. We look forward to your further feedback.

Question3:Third, the use of lipid-lowering drugs can affect patients' blood lipid bottles, and the authors did not analyze the effects of such drugs in detail in the model. At the same time, different lipid-lowering drugs will have different effects on different lipids, and the author did not conduct a detailed analysis.

Response:

Thank you for your in-depth analysis and valuable suggestions regarding our research. Concerning your point about “the use of lipid-lowering drugs can affect patients' blood lipid levels, and the authors did not analyze the effects of such drugs in detail in the model. At the same time, different lipid-lowering drugs will have different effects on different lipids, and the author did not conduct a detailed analysis,” we have given this serious consideration. We fully agree with the reviewer on the importance of the impact of lipid-lowering drugs on blood lipid levels. These drugs (such as statins, fibrates, and bile acid sequestrants) are widely used in clinical practice to treat hyperlipidemia, and they indeed have varying effects on different lipids (such as LDL, HDL, and triglycerides). Research indicates that different types of lipid-lowering drugs have varying efficacy in lowering LDL cholesterol and triglycerides, which may affect our understanding of the relationship between DI-GM and hyperlipidemia. In this study, due to the limitations of the NHANES database, we were unable to analyze in detail the use of lipid-lowering drugs and their specific effects on blood lipid levels within the model. We recognize that this limitation may affect the comprehensiveness and accuracy of the results. Thus, we will provide a detailed description of this limitation in the discussion section of the manuscript.

We fully agree with the reviewer's suggestion to include the use of lipid-lowering drugs as an important covariate in the analysis model. Future studies could consider incorporating information regarding the type, dosage, and duration of lipid-lowering drug use to more accurately assess the relationship between DI-GM and hyperlipidemia. Specifically, future research could categorize different types of lipid-lowering drugs and analyze their effects on various lipid metrics. This would provide deeper insight into the role of lipid-lowering medications in the management of hyperlipidemia. In subsequent research, we will strive to collect more comprehensive data on the use of lipid-lowering drugs, including the type, dosage, and duration of medications used by patients, to enable a more detailed analysis. We will consider employing multivariable regression models to evaluate the independent effects of DI-GM, lipid-lowering drug use, and other dietary factors on lipid levels. Such a comprehensive analysis will help reveal the interactions among different factors.

We sincerely appreciate the valuable suggestions provided by the reviewer and consider them an important direction for future research. By taking a more comprehensive approach to evaluating the impact of lipid-lowering medications, we hope to provide more valuable insights for the prevention and treatment of hyperlipidemia. We look forward to conducting further research that could yield more instructive conclusions for clinical practice, helping to improve treatment strategies for patients with hyperlipidemia. Thank you once again for your attention to our work and your recommendations; we eagerly await your further feedback.

Question4:Fourth, hyperlipidemia is a very vague clinical diagnosis, it is recommended that patients distinguish in detail between hypertriglyceridemia, hypercholesterolemia and other diseases, and analyze them separately.

Response:

Thank you for your in-depth analysis and valuable suggestions regarding our study. In response to your comment that "hyperlipidemia is a very vague clinical diagnosis, and it is recommended that patients distinguish in detail between hypertriglyceridemia, hypercholesterolemia, and other diseases, and analyze them separately," we have carefully considered this point. We fully understand that the definition of hyperlipidemia indeed contains ambiguities, typically encompassing various types of lipid abnormalities such as hypertriglyceridemia and hypercholesterolemia. Each type of hyperlipidemia may present significant differences in terms of pathological mechanisms, clinical manifestations, and treatment strategies. Therefore, a singular diagnosis of "hyperlipidemia" may not accurately reflect the specific health status of patients.In our study, we primarily focused on the relationship between DI-GM and hyperlipidemia, utilizing multiple indicators, including triglycerides, total cholesterol, LDL, and HDL, to define hyperlipidemia. Although this approach provides us with a comprehensive perspective, we recognize that the failure to adequately differentiate between the various types of hyperlipidemia may affect the interpretation of the results. Our research data is derived from the NHANES database, which offers a wealth of dietary and health information; however, due to the complexity and diversity of the data, we chose to concentrate on the impact of DI-GM in our analysis. Simultaneously, we have reviewed previous studies using the NHANES database related to hyperlipidemia, none of which fully categorized the conditions as hypertriglyceridemia or hypercholesterolemia, also focusing on the correlation between relevant indicators and hyperlipidemia. Of course, this stands as a limitation in our research. Therefore, we will provide a detailed description of this limitation in the discussion section of the manuscript.

We fully agree with the reviewer’s suggestion to analyze hyperlipidemia by categorizing it into different types. Future research could consider analyzing hypertriglyceridemia and hypercholesterolemia as independent variables to gain a clearer understanding of the pathological mechanisms behind these different types of hyperlipidemia and their specific associations with dietary factors. We recommend that future researchers adopt more detailed classification standards when designing studies to better evaluate the impact of diet on different types of hyperlipidemia.

In subsequent research, we will consider grouping hyperlipidemia patients to separately analyze the DI-GM levels of t

---

## [Decision Letter · Decision Letter 2]

16 Apr 2025

Association of the newly proposed dietary index for gut microbiota and hyperlipidemia: from the 2007-2020 NHANES study

PONE-D-24-52585R2

Dear Dr. Fang,

We’re pleased to inform you that your manuscript has been judged scientifically suitable for publication and will be formally accepted for publication once it meets all outstanding technical requirements.

Kind regards,

Pratibha V. Nerurkar, Ph.D

Academic Editor

PLOS ONE

Additional Editor Comments (optional):

Reviewers' comments:

Reviewer's Responses to Questions

**Comments to the Author**

1. If the authors have adequately addressed your comments raised in a previous round of review and you feel that this manuscript is now acceptable for publication, you may indicate that here to bypass the “Comments to the Author” section, enter your conflict of interest statement in the “Confidential to Editor” section, and submit your "Accept" recommendation.

Reviewer #1: All comments have been addressed

2. Is the manuscript technically sound, and do the data support the conclusions?

Reviewer #1: Yes

3. Has the statistical analysis been performed appropriately and rigorously? 

Reviewer #1: Yes

4. Have the authors made all data underlying the findings in their manuscript fully available?

Reviewer #1: Yes

5. Is the manuscript presented in an intelligible fashion and written in standard English?

Reviewer #1: Yes

6. Review Comments to the Author

Reviewer #1: (No Response)

7. PLOS authors have the option to publish the peer review history of their article (what does this mean? ). If published, this will include your full peer review and any attached files.

**Do you want your identity to be public for this peer review?** For information about this choice, including consent withdrawal, please see our Privacy Policy .

Reviewer #1: No

---

## [Editor Report · Acceptance letter]

PONE-D-24-52585R2

PLOS ONE

Dear Dr. Fang,

I'm pleased to inform you that your manuscript has been deemed suitable for publication in PLOS ONE. Congratulations! Your manuscript is now being handed over to our production team.

Kind regards,

on behalf of

Dr. Pratibha V. Nerurkar

Academic Editor

PLOS ONE